# Effect of Aging on the Immune Response to Core Vaccines in Senior and Geriatric Dogs

**DOI:** 10.3390/vetsci10070412

**Published:** 2023-06-23

**Authors:** Paola Dall’Ara, Stefania Lauzi, Lauretta Turin, Giulia Castaldelli, Francesco Servida, Joel Filipe

**Affiliations:** 1Department of Veterinary Medicine and Animal Sciences (DIVAS), University of Milan, Via dell’Università 6, 26900 Lodi, Italy; paola.dallara@unimi.it (P.D.); lauretta.turin@unimi.it (L.T.);; 2Clinica Veterinaria Pegaso, Via Dante Alighieri 169, 22070 Rovello Porro, Italy; francesco.servida@guest.unimi.it

**Keywords:** elderly dogs, core vaccines, canine parvovirus type 2 (CPV-2), canine distemper virus (CDV), canine adenovirus type 1 (CAdV-1), VacciCheck, antibody titration

## Abstract

**Simple Summary:**

Elderly dogs are increasingly present in households worldwide and have become routine patients in daily veterinary practice, with larger dogs aging earlier than smaller ones. Aging is not a disease but has several negative consequences on the organism in general and the immune system in particular, resulting in a decline in protection over time. Vaccines against parvovirus infection, distemper, and infectious hepatitis are highly recommended for dogs of any age, but their effectiveness in older dogs should not be assumed. The aim of this study was to measure the specific serum antibody titers against these vaccine-preventable diseases in three hundred fifty senior and geriatric dogs with the help of the rapid kit VacciCheck. More than half of the elderly population was well protected from all three diseases. Particularly, 88.6% of aging dogs were protected against parvovirus infection (in this case, larger aging dogs resulted as more protected than smaller ones), 82.3% against infectious hepatitis, and 66.0% against distemper. The advice to vaccinate on a three-year basis, properly adopted for adult dogs, should therefore be carefully considered by veterinarians for older dogs, for whom closer vaccination (every 1 to 2 years) might be more appropriate.

**Abstract:**

Elderly dogs are steadily increasing worldwide as well as veterinarians’ and owners’ interest in their health and wellness. Aging is not a disease, but a combination of changes negatively affecting the organism in general and the immune system in particular, resulting in a decline in protection over time. The aim of this study was to measure the specific serum antibody titers against the main dangerous and widespread viral diseases preventable by core vaccinations in senior and geriatric dogs using the in-practice test VacciCheck. A cohort of three hundred fifty elderly dogs was analyzed for Protective Antibody Titers (PATs) against CPV-2, CDV and CAdV-1. The age ranged from 5 to 19 years, with two hundred fifty-eight seniors (73.7%) and ninety-two geriatrics (26.3%), and 97.4% of them were vaccinated at least once in their lives. More than half of the entire study population (52.9%) had PATs simultaneously for all three diseases, with 80.5% seniors and 19.5% geriatrics. Specific PATs were found in 88.6% of aging dogs for CPV-2, 82.3% for CadV-1 and 66.0% for CDV, demonstrating that unprotected aging dogs represent a minority. Unexpectedly, the larger elderly dogs resulted as more protected than smaller ones for CPV-2. Protection then decreases over time, with geriatric dogs less protected than senior ones. Veterinary practitioners should therefore always consider whether to maintain core vaccinations in aging dogs as in adults on a three-year basis or opt instead for closer boosters (every 1 or 2 years). PATs for core vaccines could then represent a good biomarker of protection and their titration could become a standard of care, especially in such a sensitive period of the dogs’ life.

## 1. Introduction

In veterinary practices, senior dogs account nowadays for 40–50% of canine patients, with both of their percentages and average ages steadily increasing, resulting in a growing focus on their health and wellness [1,2,3,4,5]. However, when is a dog considered old? The biological age correlates with the chronological one but is not synonymous with it. The latest (2023) guidelines of the American Animal Hospital Association (AAHA) report the recommendations of a task force of experts with decades of collective experience in the care of elderly patients [1] and define as “senior” an older aging pet; however, given the great variability in aging by breed and size, there is no a specific age in defining the senior status. This is especially true in dogs, where there is clear evidence that longevity is inversely related to size (both weight and height); larger dogs not only show the typical negative consequences of aging earlier than smaller ones, but also die younger [6,7,8,9]. Finally, a very recent paper by Montoya et al. [10], which aimed at calculating the life expectancies (LE) of dogs, reported a canine average lifespan of 12.69 years, confirming that larger dogs will age earlier (giant LE 9.51 years, large LE 11.51 years) than smaller dogs (medium LE 12.7, small LE 13.53, toy LE 13.36 years). Accordingly, veterinarians should consider dogs in need of elderly care at differing times depending on their size, starting from 5–6 to 9 years old for giant/large breeds and from 9 to 13 years old for medium/small breeds (Table 1) [11,12,13].

Aging is not a disease, but a combination of physical and dynamic changes in the organism starting from conception, continuing with adulthood and senescence, and ending with death [14,15]. In this process, there are many different factors influencing the aging rate, generating alteration in musculoskeletal, adipose, and cerebral tissue and leading to physical and functional phenotypical changes typical of older patients [16]. These modifications also affect the immune system. Immunosenescence is in fact a dynamic process of immune dysfunction that occurs with age involving a continuous remodeling of lymphoid organs and immune cells, in which some immune parameters increase while others decrease or remain unchanged [17]. These changes undeniably lead to a greater susceptibility to the development of the so-called Age-Related Diseases (ARDs), particularly infectious, autoimmune, and neoplastic diseases [17,18,19,20,21]. While evidence of this phenomenon has been well reported in human medicine, for a long time less was known about dogs, as immunosenescence was one of the less studied and understood areas of veterinary geriatric medicine. In the last 15 years, however, something has changed [7,15,16,22,23,24,25,26]. In addition, the peculiar term “inflammaging”, now used in both human and veterinary medicine, has been coined to describe the effects of a continuous antigenic challenge throughout life leading to the production of inflammatory cytokines triggering the onset of inflammatory conditions in later life [1,7,20,21,27].

Although both cell-mediated and humoral immunity are weakened during the aging process, in both human and veterinary medicine the cell-mediated compartment is the most affected. This explains why elderly patients are more susceptible to infectious diseases and neoplasms than younger ones [7,15,25]. This decline in cell-mediated function is considered a result of the thymic involution. Thymus is a central primary lymphoid organ responsible for the production of T lymphocytes, which represent the pivotal cells in orchestrating both types of immunity (cell-mediated and humoral ones). After birth, the thymus continues its development until puberty, and at that time (in dogs generally at 4–5 months of age) it starts a slow but progressive involution, leading to a significant decrease in production and activity of thymic hormones and T lymphocytes (helper and cytotoxic ones) [15,19,20,28]. This unavoidable thymus involution is regarded as one of the main factors contributing to the loss of immune function typical of the elderly and is considered in turn a genetically programmed event (the so-called “thymus clock”) [15,17,21,29]. In a recent study (2022), Lee et al. proposed both CD4/CD8 ratio and Growth Differentiation Factor 8 (GDF8) levels in canine blood as useful factors to use as an age prediction model for dogs, as these parameters were found to be significantly correlated with age [30].

Clinical evidence in humans and animals suggest that, as age advances, the secondary (anamnestic) immune response against known antigens (for example vaccine boosters) still works well, whereas the ability to mount a primary immune response against new antigens never encountered before (field or vaccinal ones) declines significantly with age for different causes, always leading to a dangerous increased susceptibility to infectious diseases, which may also limit the effectiveness of vaccination in older individuals [15,16,17,18,22,26,31,32]. Few studies, however, have been conducted to date to understand if these age-related changes in the elderly canine patients are able to affect the immune response to vaccines.

Canine and feline vaccines are classified into core and non-core by all the international vaccination guidelines [33,34,35,36,37]. Core vaccines are the recommended ones, i.e., the ones that every dog and every cat should receive almost once in their life, as they are intended for widespread, dangerous, contagious, and even life-threatening diseases. The canine core vaccines protect dogs against canine parvovirus infection (CPV-2), canine distemper (CDV) and infectious canine hepatitis (CAdV-1) [33,34,38]. It has been shown that any dog can be infected regardless of age, and cases of core diseases are reported also in senior dogs [39,40,41]. For this reason, core vaccinations are considered the main tool to control these diseases. For adult dogs the international guidelines recommend core revaccination on a three-year basis using modified live vaccines (MLV), whereas for elderly canine patients only a few indications are available. Indeed, many associations, researchers, and veterinary practitioners have developed detailed canine and feline senior and geriatric care guidelines in order to focus on the main issues regarding elderly patients, but they rarely refer to a particular vaccination protocol for this delicate phase of life or to the need to adapt those already existing for adult pets [1,2,3,5,42,43,44].

The aim of this study was to measure the specific serum antibody titers, and therefore the actual protection, against CPV-2, CDV and CAdV-1 in senior and geriatric dogs by using an in-practice test kit.

## 2. Materials and Methods

### 2.1. Study Population and Study Protocol

Plasma/serum samples analyzed in this study were collected over 9 years (January 2014 to January 2023) to analyze specific antibody titers for core diseases and for other purposes (e.g., diagnostic assays, surgery, health checks) related to patient aging. According to the Ethical Committee position of the University of Milan, residual aliquots of biological samples that were collected under the owner’s informed consent can be freely used for research purposes without any formal approval (EC decision 29 October 2012, renewed with the protocol n. 02-2016). For each patient, different key information was recorded: (1) age: seniors and geriatrics (the age category was determined according to the size of the dogs, following the aforementioned rule that larger dogs age earlier than smaller ones and vice versa [8,9,10]); (2) breed (purebred or crossbred) and breed size: small (<10 kg), medium (≥10–<25 kg), and large (≥25 kg); (3) sex and reproductive status: intact or neutered male or female; (4) health status: healthy or unhealthy, considering as unhealthy only animals with clinical problems that could affect immune function (none of the unhealthy dogs were, however, being treated with drugs that might interfere with the immune system (neither immunosuppressive nor immunostimulant): for example, none of the oncological patients was under chemotherapy) (see below Results section); (5) vaccination history, considering the time elapsed since the last core vaccination: ≤1 year, >1 year–≤3 years, or >3 years.

### 2.2. Detection of Specific Antibodies to Core Vaccines by VacciCheck and Expression of the Results

Each plasma/serum sample was tested with the in-clinic test Canine VacciCheck^®^ (produced by Biogal, Kibbutz Galed, Israel, and supplied in Italy by Agrolabo, Scarmagno, Italy) following the manufacturer’s instruction. VacciCheck^®^ is a rapid semiquantitative dot-ELISA-based kit able to determine specific antibody titers against CPV-2, CDV, and CAdV-1 in serum, plasma, or whole blood of dogs. This test has good specificity and sensitivity for each virus and can be very useful in daily veterinary practice to evaluate the real protection of dogs (and cats) against diseases preventable by core vaccines, as suggested by the international vaccination guidelines and by many other studies [1,33,45,46,47,48,49]. In this test, the antibody concentration is directly proportional to the color intensity of the related spots read by the CombScale and expressed as “S” units on a scale from 1 to 6. The S value of 0 (S0) is equivalent to an antibody titer <1:20 for CPV-2, <1:8 for CDV, and <1:4 for CAdV-1 and is considered negative, while the S value of 3 (S3) is equivalent of the threshold values of 1:80 for CPV-2, 1:32 for CDV, and 1:16 for CAdV-1. Antibody titers equal to or higher than S3 are considered indicative of a significant positive response, corresponding to a good protection against these three core diseases (Appendix A). In this study, results were divided into four categories (unprotected, weak positive, medium positive, and high positive) based on the threshold values of each pathogen, and dogs with antibody titers equal to the different threshold values were considered medium positive. Medium to high positive results were expressed as Protective Antibody Titers (PATs) (Appendix A), as already done in our recent previous study [50] following the suggestions of the literature [22,51,52,53,54,55,56,57,58].

### 2.3. Statistical Analysis

Statistical analyses were carried out using GraphPad Prism 9 (La Jolla, CA, USA), values at *p*-value < 0.05 were considered statistically significant, and those with *p*-value < 0.1 as a tendence. In order to determine significant differences between protected and unprotected vaccinated dogs, the Chi-square (χ^2^) analysis was performed. All antibody titer data were transformed with log_2_. The Shapiro–Wilk test was used to verify the normal distribution of data, together with the not-parametric Kruskal–Wallis and Mann–Whitney tests.

## 3. Results

### 3.1. Dog Population

A total of three hundred fifty plasma/serum samples of owned older dogs were included in the study. The age ranged from 5 to 19 years, with two hundred fifty-eight seniors (73.7%) and ninety-two geriatrics (26.3%). Considering the breed, two hundred thirty-six were purebred (67.4%) belonging to sixty-four different breeds, while one hundred fourteen were crossbred (32.6%); the most representative breeds were Golden Retriever (18, 5.1%), Labrador Retriever (16, 4.65%), Poodle and Beagle (both 13, 3.7%), and Cocker Spaniel (12, 3.4%). Regarding their size, one hundred twenty-four were small (35.4%), one hundred twenty-three medium (35.1%) and one hundred and three large (29.4%). Considering sex and reproductive status, one hundred eighty-four (52.6%) were females (eighty-nine sexually intact and ninety-five neutered) and one hundred sixty-six (47.4%) were males (one hundred thirty-two sexually intact and thirty-four neutered). Collectively, two hundred twenty-one dogs (63.1%) were intact, while one hundred twenty-nine (36.9%) were neutered. Regarding their health status, two hundred sixty dogs were healthy (74.3%), while ninety (25.7%) had one or more clinical problems including neoplasms (63, 18.0%), endocrinopathies (13, 3.7%), heart problems (7, 2.0%), and others (7, 2.0%), many of which could impact immune function. Finally, for all three hundred fifty patients the vaccination history was known: three hundred forty-one (97.4%) dogs were vaccinated at least once in their lives, while nine (2.6%) had never been vaccinated in their lifetimes. Regarding time elapsed from the last vaccine injection, one hundred fifteen out of the three hundred forty-one vaccinated older patients were vaccinated ≤1 year before sampling (33.7%), one hundred thirty-four received their last vaccination >1–≤3 years earlier (39.3%), and the remaining ninety-two were vaccinated more than 3 years earlier (27.0%).

### 3.2. Antibody Titers and Protection of the Aging Dogs

Antibody titers specific for the three diseases ranged from <1:20 to >1:640 for CPV-2, from <1:8 to >1:256 for CDV, and from <1:4 to >1:128 for CAdV-1. Specific PATs for CPV-2, CDV, and CAdV-1 were found in three hundred and ten (88.6%), two hundred thirty-one (66.0%) and two hundred eighty-eight (82.3%) out of the three hundred fifty older dogs, respectively. Percentages were a little bit higher considering only the three hundred forty-one vaccinated dogs (89.4%, 67,4% and 83.9%, respectively). These percentages were significantly higher if titers just below the threshold values (weak positive) were also considered: 95.7% CPV-2 (335/350 dogs), 89.4% CDV (313/350 dogs), and 95.4% CAdV-1 (334/350 dogs) showing that unprotected aging dogs represent a minority (4.3%, 10.6 and 4.6%, respectively).

PATs found in the elderly dog population both overall and divided into categories (age, size, sex and reproductive status, health status, and time since last vaccination when applicable) are shown in Table 2, Figure 1 and Appendix A.

Results of the Chi square test applied to three hundred fifty dogs are reported in Table 3 (in bold red statistically significant *p*-values are highlighted). Figure 2, Figure 3, Figure 4 and Figure 5 show the statistical results related to the three hundred fifty dogs divided into the different categories.

Finally, results of the nine unvaccinated older dogs are reported in Table 4.

Dogs with PATs simultaneously for all the three diseases were found to be more than half of the entire study population (one hundred eighty-five out of three hundred fifty, 52.9%): of these, one hundred forty-nine were seniors (80.5%) and thirty-six were geriatrics (19.5%). On the contrary, dogs resulting as unprotected for all pathogens simultaneously were sixteen out of three hundred fifty (4.6% of the population), of which twelve were seniors (75%) and four were geriatrics (25%).

### 3.3. Results According to the Different Variables

#### 3.3.1. Age

Comparing elderly dogs, geriatric dogs had statistically lower antibody titers against CDV than senior ones resulting in lower protection both in terms of percentages (56.5% vs. 69.4%, *p* = 0.0254) and PAT values (*p*-value = 0.0295) (Figure 2). In addition, CPV-2 geriatric dogs tended to be less protected than senior ones (83.7% vs. 90.3%, *p* = 0.0869). The same difference was not seen for CAdV-1 (80.4% vs. 82.9%, *p* = 0.5881) (Table 2).

**Figure 2 vetsci-10-00412-f002:**
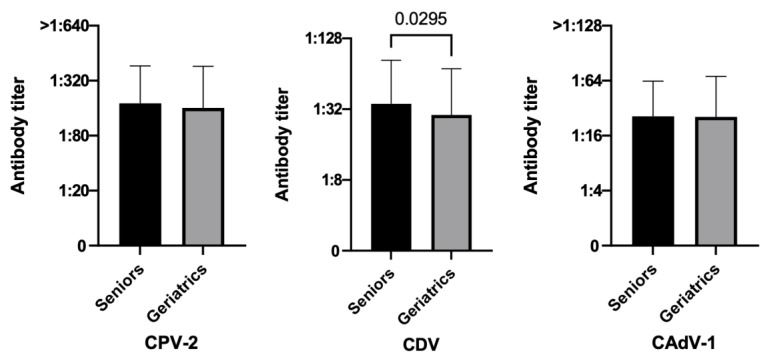
Antibody titers against Canine Parvovirus type 2 (CPV-2), Canine Distemper Virus (CDV), and Canine Adenovirus type 1 (CAdV-1), considering the variable ages of the three hundred fifty dogs (Mann–Whitney test).

#### 3.3.2. Size

Considering the variable breed size positivity rates for all three diseases increased as the size raised, and larger elderly dogs tended to be more protected than smaller ones for CPV-2 both in terms of percentages (94.2% large dogs vs. 87.8% medium dogs and 84.7% small dogs, *p* = 0.0772) (Table 2), and PAT values, the latter in a statistically significant way (large vs. small *p =* 0.0041, large vs. medium *p =* 0.0353) (Figure 3).

**Figure 3 vetsci-10-00412-f003:**
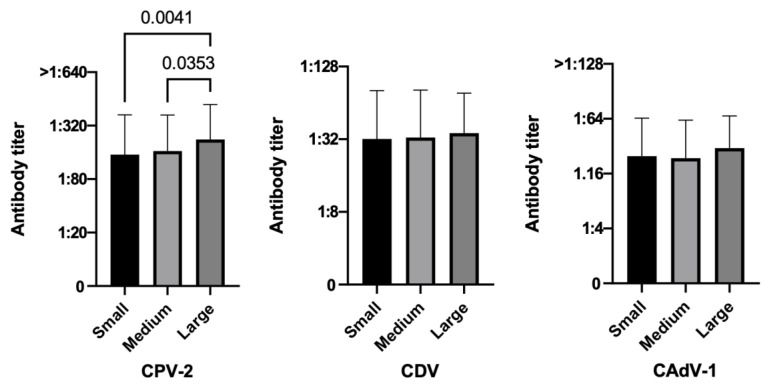
Antibody titers against Canine Parvovirus type 2 (CPV-2), Canine Distemper Virus (CDV), and Canine Adenovirus type 1 (CAdV-1), considering the variable sizes of the three hundred fifty dogs (Kruskal–Wallis test).

#### 3.3.3. Sex and Reproductive Status

Regarding sex and reproductive status, neutered females and intact males had the highest number of protected animals for all the three core vaccines, but no differences were statistically significant except CAdV-1 (*p* = 0.0013), probably due to the higher percentage of unprotected intact females (31.5%) (Table 2). However, no statistical differences were found when comparing PAT values.

#### 3.3.4. Health Status

Considering health status, healthy dogs tended to have a greater number of protected subjects than unhealthy ones for CPV-2 in terms of percentages (90.4% vs. 83.3%, *p* = 0.0835) (Table 2), and in a statistically significant way of PAT values (*p =* 0.0153) (Figure 4).

**Figure 4 vetsci-10-00412-f004:**
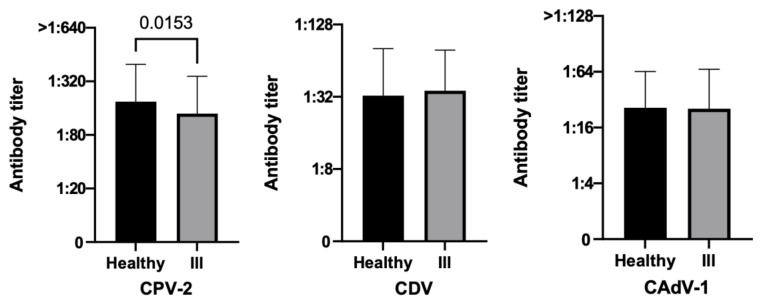
Antibody titers against Canine Parvovirus type 2 (CPV-2), Canine Distemper Virus (CDV), and Canine Adenovirus type 1 (CAdV-1), considering the variable health statuses of the three hundred fifty dogs (Mann–Whitney test).

#### 3.3.5. Time Elapsed since the Last Vaccination

As time passed, vaccine protection has shown a significant decline in the number of protected dogs for CDV (*p* = 0.0007) and CAdV-1 (*p* < 0.0001) (Table 2). Moreover, dogs vaccinated less than 1 year before were more protected than dogs vaccinated more than 3 years before, and these differences were statistically significant for all the three diseases in terms of PAT values (CPV-2 *p* = 0.0002, CDV *p* = 0.0003, CAdV-1 *p* < 0.0001) (Figure 5).

**Figure 5 vetsci-10-00412-f005:**
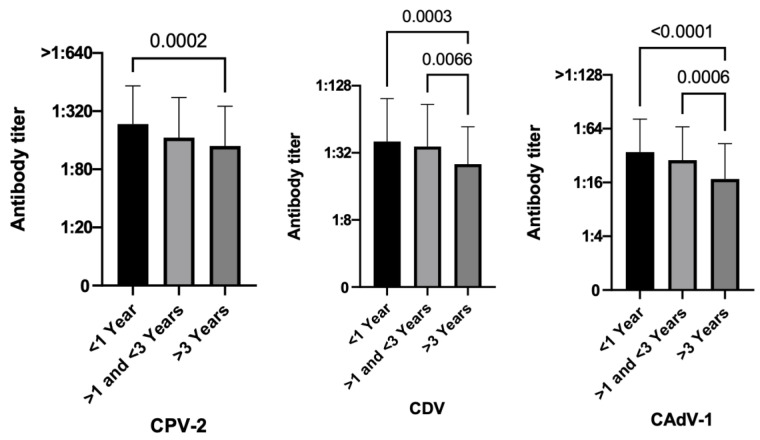
Antibody titers against Canine Parvovirus type 2 (CPV-2), Canine Distemper Virus (CDV), and Canine Adenovirus type 1 (CAdV-1), considering the variable time elapsed since the last vaccinations of the three hundred fifty dogs (Kruskal–Wallis test).

## 4. Discussion

Senior and geriatric pets increasingly represent a reality with which veterinary practitioners worldwide have to deal with daily. Therefore, knowing their immune responses against the most important vaccine-preventable diseases is a great help in their management given the almost total lack of precise indications in the scientific literature.

The results of this study are comforting since they were in line with those of our previous work on a cohort of one thousand and twenty-seven Italian dogs of different ages showing 90.8%, 68.6%, and 79.8% protection against CPV-2, CDV, and CAdV-1, respectively [51]. In fact, specific PATs were found in the majority of elderly dogs, with percentages significantly increasing when the weak positive dogs were considered, which demonstrated that just a few aging dogs were not protected.

CPV-2 was confirmed as the pathogen with the highest values both in terms of the number of protected subjects (310/350, 88.6%) and protective effects (1 or 2 titers higher than the threshold value, Appendix A). This supremacy was likely related to CPV-2 extensive field circulation in rural and urban contexts, and to its well-known environmental resistance and its strong immunogenicity. All of these reasons can explain both the natural booster effect of CPV-2 in vaccinated dogs with high rates of seroprotection, also in the elderly canine population [50,51], and, vice versa, cases of disease in adult and older dogs (sometimes even vaccinated) [39,59]. It should also be considered that cats can be infected with CPV-2 and can shed the virus in the environment with their feces, thus posing a real though uncommon risk to dogs [60,61,62].

The second highest rate of positivity was related to canine infectious hepatitis (288/350, 82.3%), a value that was 2.5 percentage points higher than those of the previous study on a cohort of Italian dogs (79.8%) [51]. In the literature, a higher incidence of CAdV-1 infection has been reported in elderly dogs aged over 10 years [63]. Moreover, in this case the high values of CAdV-1 PATs can be related to their environmental resistance; however, circulation in Italy of this virus is restricted to wild animals and some dogs, especially stray ones in the Southern regions, and the last clinical manifestations of the disease in Italian dogs date back many years [64,65]. Moreover, the presence of CAdV-1 PATs could strengthen the hypothesis of a cross-reaction between CAdV-1 and CAdV-2, pathogens equally resistant in the environment for weeks or even months affecting the canine respiratory tract, circulating in dogs’ populations and spread from dog to dog through coughing and present in canine hepatitis vaccines instead of CAdV-1 due to the adverse effects of the latter [33,66,67].

The lowest positivity rate was associated with CDV (231/350, 66.0%). This was a predictable result since this virus is labile in the environment and its immunogenicity is lower than that of the previous two viruses and even with successful vaccination it tends to develop lower titers if compared with CPV. Moreover, it has immunosuppressive potential and post-vaccination antibody titers are lower than those of CPV-2 [49,67,68].

More than half of the elderly dog population in this study were found to be protected for all three core diseases, while only sixteen (4%) were unprotected for all. Among them, three dogs had never been vaccinated in their lives, and seven dogs had been vaccinated more than 3 years before. These two conditions probably accounted for the total lack of protection found in those ten dogs. On the other hand, the remaining six unprotected dogs were vaccinated more recently (from 1 month to 2.5 years before). This is a critical point to take into serious consideration when planning the vaccination protocol for an elderly dog.

Finally, none of the nine unvaccinated dogs had PATs for all three diseases, neither seniors nor geriatrics, but five out of nine (55.5%) resulted as protected against the most dangerous one (CPV-2), and three out of the remaining four had CPV-2 titers just below the threshold value (1:40). These results confirmed the circulation of this pathogen in the environment and its ability to stimulate a specific immune response without always causing disease but only in subclinical forms, as previously reported [69,70].

Considering the age variable, geriatric dogs were found to have significantly lower protection for distemper compared to senior dogs (*p*-value = 0.00295), and a trend to significance (*p*-value = 0.0869) was found for parvovirus infection when comparing protected and unprotected animals. However, no differences were found for infectious hepatitis. These results were confirmed by analyzing antibody responses by time since the last vaccination, as protection clearly decreases over time. A decrease in serum antibody levels over time is a well-known event reported by many authors, but generally this decline is not significant, since immunity to core vaccines can persist for life after vaccination, in a manner quite similar to what happens after a natural infection [7,24,71]. These results seem to confirm the indications of the WSAVA guidelines and of other experts [24,33,71], which stress that any antibody titer found in an adult (in this case, even an elderly dog) is a marker of protection because it indicates the presence of an underlying immune memory. In fact, experts underlie that old dogs can die from a core vaccine-preventable disease only if they have never been vaccinated [24], since immunization failures, above all for the most dangerous and widespread pathogen (CPV-2), are uncommon today [59]. This difference between senior and geriatric dogs, however, should give pause for thought about the appropriateness of maintaining a three-year basis vaccination protocol, strongly recommended for adult and senior dogs and cats, to geriatric individuals as well, since this vaccination scheme could not confer protection to older dogs [22,26]. In 2016, Ellis et al. [22] compared the serological responses to different vaccinal antigens (and among them the two core CPV-2 and CDV) in lean and obese geriatric dogs, demonstrating that not only age but also nutritional status could affect the vaccine response, with lean and obese geriatric dogs having lower protection to CPV-2 and CDV than young normal-weight dogs, making this another aspect that should be taken into serious consideration when planning a vaccination protocol for older dogs.

Unexpectedly, the larger aging dogs of this study resulted as more protected than smaller ones, and for CPV-2 this difference resulted statistically significant (*p*-value = 0.0041). This result is not in line with other studies in which smaller dogs better respond to antigenic stimulation and vaccination compared to larger sized dogs [38,58,72,73]. Furthermore, in our recent study on a cohort of one thousand and twenty-seven Italian dogs [50], larger dogs have been generally considered less protected than smaller ones. Nevertheless, considering only CPV-2 protection, the peculiarity of more protected larger dogs has been already reported, especially when considering only the nine hundred fifty-one vaccinated animals. CPV-2 is certainly one of the most known immunogenic viruses, which can strongly stimulate a specific immune response in animals with which it comes in contact (for field or vaccine infections). It is also widely distributed in the environment where it is excreted by sick dogs or healthy carriers (and also by infected cats [60,61,62,74,75]) through their feces, and it can persist for very long periods due to its extreme environmental resistance [67,69,70,76,77]. Furthermore, it is closely antigenically and genetically related to Feline Panleukopenia Virus (FPV) (approx. 98% identity [78]), an equivalently resistant and widespread virus [79]. Larger dogs lead less of a home life and are not ideal for small apartments, while they are considered excellent outdoor companions and need more exercise than smaller ones [80]. Furthermore, many large dogs are kept outside (despite the fact that their owners consider them companions and not working dogs) or have an indoor/outdoor life, and many are allowed to roam free going outside with their owners [81]. Finally, larger breeds are generally chosen by owners who prefer to spend time in outside activities [82], and take their dogs outside more often and for longer periods than owners of smaller dogs. Smaller dogs sometimes live exclusively indoors or go outside but in their owners’ arms or bags, and for these reasons the probability of coming into contact with CPV-2 or FPV might be higher for larger dogs.

Regarding sex and comparing protected with unprotected animals, neutered females and intact males showed higher percentages of covered individuals for all of the diseases. On the contrary, intact females were the group with lower protection for infectious hepatitis (result extrapolated from a Chi-square test, *p*-value = 0.0013). This last result agrees with other studies where sterilization is reported to stimulate immune responses [73], and partially also confirms those of our previous study [50] where neutered dogs have been reported to have higher antibody titers compared with intact ones.

Finally, and as expected, healthy older dogs were significantly more and better protected than unhealthy dogs for CPV2 (*p*-value = 0.00153), while for the other two diseases there were no significant differences. It is likely that the diseases carried by the dogs in this study, which were considered unhealthy (especially neoplasms, and among these lymphomas and mastocytomas, and endocrinopathies), did not have a major impact on the immune system and the response to previous vaccination was as might have been expected, probably because they were either at an early stage or kept under control by proper therapies.

This study was able to demonstrate that with aging the specific immune response towards core vaccines undergoes a physiological decline in elderly dogs, but remains at protective levels for most subjects. In fact, dogs vaccinated less than one year before are more protected than those vaccinated than three years before for all three diseases. This study should then make veterinary practitioners think about the appropriateness to maintain core vaccinations on a three-year basis with aging and not opt instead for closer boosters (every 1 or 2 years) in order not to leave already fragile subjects without protection. Further studies on a larger canine cohort will be needed to better elucidate the peculiar aspects of the vaccine immune response in older dogs highlighted in this work, first of all, the high response of larger dogs against CPV-2. Given the known good correlation between antibody titers and protection suggested by the WSAVA guidelines on vaccination of dogs and cats [33], it is possible however to state that the antibody titers for core vaccines represent a good biomarker of protection. Antibody titration (e.g., with VacciCheck) may be a very useful tool in monitoring canine immunity and could become a standard of care in such a delicate period of the dogs’ life. It would also be interesting to extend the study to older cats to confirm whether the specific immunity for core vaccines remains high also in the feline species despite aging or whether it would be necessary to adapt the vaccination protocols to elderly cats in order to avoid dangerous vaccine failures.

This study has some limitations. First of all, the vaccination protocols chosen and applied for each elderly patient was unknown, since only the date of the last core vaccination was available. Therefore, it cannot be determined whether the good protection evidenced in the elderly dogs of this study is the result of a continuous veterinary approach with vaccinations that are too frequent (i.e., on an annual basis and not on a three-year basis despite guidelines and experts’ indications) or whether it is actually related to the excellent stimulation of the immune system by MLV vaccines administered on a triennial basis during the entire dogs’ life. Moreover, not knowing other details of the lives of these elderly patients (e.g., nutritional status, lifestyle, frequency of veterinary visits, etc.) is another limitation of this study, as this additional information would have helped to better understand and interpret the obtained results. The evaluation of specific vaccine immune response is also incomplete, as it is known that it can stimulate also a strong but not easily measurable cell-mediated immunity [50,51,83]. In elderly subjects, however, the aging process particularly affects cell-mediated immunity [7,15,17,19,20,25,28,29], so adding information related to this type of immune response would not add other useful information. One can argue that the sample size of this study was not adequate, however we believe that the sample we analyzed can be considered representative enough of the elderly canine population in Italy and can be helpful to veterinary practitioners in appropriately managing the increasing number of elderly patients they have to visit and provide care for on a daily basis.

## 5. Conclusions

The number of elderly dogs is steadily growing worldwide, now accounting for half of all veterinary pet patients. It is therefore necessary for veterinarians and owners to work together to maintain a high quality of life for their patients and pets by understanding the biological effects of aging and implementing preventive and therapeutic actions to safeguard their delicate health.

Taken together, the results of this study underlie the importance of optimizing vaccination programs and controlling protection in aging dogs, especially in geriatric ones, in which immunosenescence could hamper mounting an adequate immune response after vaccination, increasing the risk of developing dangerous and even life-threatening infectious diseases. Moreover, given the known difficulty of an elderly individual to mount an immune response toward a new antigen never encountered before, the “trick” to try to ensure vaccine protection even in elderly pets is to make the immune system not lose its memory. This would be possible for continuing vaccinations also in the elderly stages, while explaining this rationale to owners who can feel that their dogs are too old for vaccinations.

## Figures and Tables

**Figure 1 vetsci-10-00412-f001:**
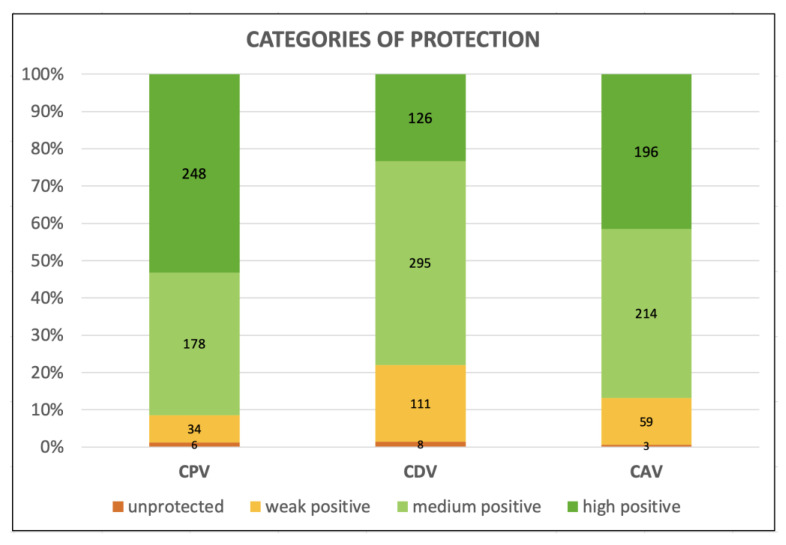
Categories of protection against Canine Parvovirus type 2 (CPV-2), Canine Distemper Virus (CDV), and Canine Adenovirus type 1 (CAdV-1) in the three hundred fifty aging dogs.

**Table 1 vetsci-10-00412-t001:** Dogs’ physiological ages in human years based on their sizes [12].

	DOGS’ PHYSIOLOGICAL AGES IN HUMAN YEARS			
	DOGS’ SIZES			
AGE (Years)	≤9 kg	10–22 kg	23–41 kg	>41 kg
**5**	--	--	--	42
**6**	--	--	45	49
**7**	44	47	50	56
**8**	48	51	55	64
**9**	52	56	61	71
**10**	56	60	66	78
**11**	60	65	72	86
**12**	64	69	77	93
**13**	68	74	82	101
**14**	72	78	88	108
**15**	76	83	93	115
**16**	80	87	99	123
**17**	84	92	104	--
**18**	88	96	109	--
**19**	92	101	115	--
**20**	96	105	120	--

Green: Senior dogs–Blue: Geriatric dogs.

**Table 2 vetsci-10-00412-t002:** Percentages and numbers (*in italics* in brackets) of senior and geriatric dogs with Protective Antibody Titers (PATs) for Canine Parvovirus type 2 (CPV-2), Canine Distemper Virus (CDV), and Canine Adenovirus type 1 (CAdV-1) according to age, size, sex and reproductive status, and health status of the entire elderly dog cohort (three hundred fifty dogs), and also according to the time elapsed since the last core vaccination for the three hundred forty-one vaccinated dogs.

	Protective Antibody Titers (PATs)—% (*n. of Dogs*)		
	CPV-2	CDV	CAdV-1
*Threshold values*	*1:80*	*1:32*	*1:16*
**Protected dogs (with PATs)**	**88.6** (*310/350*)	**66.0** (*231/350*)	**82.3** (*288/350*)
**Unprotected dogs (under PATs)**	**11.4** (*40/350*)	**34.0** (*119/350*)	**17.7** (*62/350*)
**Age**			
Seniors (*258*)	**90.3** (*233/258*)	**69.4** (*179/258*)	**82.9** (*214/258*)
Geriatrics (*92*)	**83.7** (*77/92*)	**56.5** (*52/92*)	**80.4** (*74/92*)
**Size**			
Small (<10 kg) (124)	**84.7** (*105/124*)	**61.3** (*76/124*)	**80.6** (*100/124)*
Medium (≥10–≤25 kg) (123)	**87.8** (*108/123*)	**65.0** (*80/123*)	**78.9** (*97/123*)
Large (>25 kg) (103)	**94.2** (*97/103*)	**72.8** (*75/103*)	**88.3** (*91/103*)
**Sex and reproductive status**			
Intact females (89)	**86.5** (*77/89*)	**58.4** (*52/89*)	**68.5** (*61/89*)
Neutered females (95)	**87.4** (*83/95*)	**67.4** (*64/95*)	**86.3** (*82/95*)
Intact males (132)	**90.2** (*119/132*)	**69.7** (*92/132*)	**87.9** (*116/132*)
Neutered males (34)	**91.2** (*31/34*)	**67.6** (*23/34*)	**85.3** (*29/34*)
**Health status**			
Healthy (260)	**90.4** (*235/260*)	**64.2** (*167/260*)	**82.3** (*214/260*)
Unhealthy (90)	**83.3** (*75/90*)	**71.1** (*64/90*)	**82.2** (*74/90*)
**Time after vaccination ***			
≤1 year (115)	**92.2** (*106/115*)	**76.5** (*88/115*)	**90.4** (*104/115*)
>1–≤3 years (134)	**88.8** (*119/134*)	**70.1** (*94/134*)	**90.3** (*121/134*)
>3 years (92)	**87.0** (*80/92*)	**52.2** (*48/92*)	**66.3** (*61/92*)

* These percentages were calculated for the three hundred forty-one vaccinated dogs out of the total three hundred fifty.

**Table 3 vetsci-10-00412-t003:** Percentages and numbers (in italics in brackets) of the Chi square test for Canine Parvovirus type 2 (CPV-2), Canine Distemper Virus (CDV), and Canine Adenovirus type 1 (CAdV-1) antibody protection according to age, size, sex and reproductive status, health status, and time elapsed since the last vaccination of the three hundred forty-one vaccinated senior and geriatric dogs.

		CPV-2			CDV			CAdV-1		
		PROTECTED		*p*-Value	PROTECTED		*p*-Value	PROTECTED		*p*-Value
Statistical Variable (*Number*)		YES	NO		YES	NO		YES	NO	
**Age**	Seniors (*258*)	90.3% (*233*)	9.7% (*25*)	0.0869	69.4% (*179*)	30.6% (*79*)	** 0.0254 **	82.9% (*214*)	17.1% (*44*)	0.5881
	Geriatrics (*92*)	83.7% (*77*)	16.3% (*15*)		56.5% (*52*)	43.5% (*40*)		80.4% (*74*)	19.6% (*18*)	
**Size**	Small (*124*)	84.7% (*105*)	15.3% (*19*)	0.0772	61.3% (*76*)	38.7% (*48*)	0.1819	80.6% (*100*)	19.4% (*24*)	0.1483
	Medium (*123*)	87.8% (*108*)	12.2% (*15*)		65.0% (*80*)	35.0% (*43*)		78.9% (*97*)	21.1% (*26*)	
	Large (*103*)	94.2% (*97*)	5.8% (*6*)		72.8% (*75*)	27.2% (*28*)		88.3% (*91*)	11.7% (*12*)	
**Sex**	Intact F (*89*)	86.5% (*77*)	13.5% (*12*)	0.7866	58.4% (*52*)	41.6% (*37*)	0.3620	68.5% (*61*)	31.5% (*28*)	** 0.0013 **
	Neutered F (*95*)	87.4% (*83*)	12.6% (*12*)		67.4% (*64*)	32.7% (*31*)		86.3% (*82*)	13.7% (*13*)	
	Intact M (*132*)	90.2% (*119*)	9.8% (*13*)		69.7% (*92*)	30.3% (*40*)		87.9% (*116*)	12.1% (*16*)	
	Neutered M (*34*)	91.2% (*31*)	8.8% (3)		67.6% (*23*)	32.4% (*11*)		85.3% (*29*)	14.7% (*5*)	
**Health** **status**	Healthy (*260*)	90.4% (*235*)	9.6% (25)	0.0835	64.2% (*167*)	35.8% (*93*)	0.2482	82.3% (*214*)	17.7% (*46*)	0.9999
	Unhealthy (*90*)	83.3% (*75*)	16.7% (*15*)		71.1% (*64*)	28.9% (*26*)		82.2% (*74*)	17.8% (*16*)	
**Time after vaccination**	≤1 year (*115*)	92.2% (*106*)	7.8% (*9*)	0.4565	76.5% (*88*)	23.5% (*27*)	** 0.0007 **	90.4% (*104*)	9.6% (*11*)	** <0.0001 **
	>1–≤3 years (*134*)	88.8% (*119*)	11.2% (*15*)		70.1% (*94*)	29.9% (*40*)		90.3% (*121*)	9.7% (*13*)	
	>3 years (*92*)	87.0% (*80*)	13.0% (*12*)		52.2% (*48*)	47.8% (*44*)		66.3% (*61*)	33.7% (*31*)	

In **bold red**, statistically significant *p*-values–F = female; M = male.

**Table 4 vetsci-10-00412-t004:** Antibody titers for Canine Parvovirus type 2 (CPV-2), Canine Distemper Virus (CDV), and Canine Adenovirus type 1 (CAdV-1) and features of the nine unvaccinated senior (4) and geriatric (5) dogs (in green Protective Antibody Titers, PATs).

SENIORS							
Age (Years)	Breed	Size	Sex	Health	CPV-2 *1:80*	CDV *1:32*	CAdV-1 *1:16*
8	Beagle	Medium	F	Unhealthy	40	16	8
9	American Bulldog	Large	F	Healthy	160	16	8
8	Crossbred	Small	NM	Unhealthy	80	16	8
10.5	Crossbred	Small	F	Healthy	320	8	8
**GERIATRICS**							
**Age (Years)**	**Breed**	**Size**	**Sex**	**Health**	**CPV-2** ** *1:80* **	**CDV** ** *1:32* **	**CAdV-1** ** *1:16* **
12	Crossbred	Medium	M	Healthy	160	16	8
15	Cocker Spaniel	Medium	NF	Healthy	40	8	4
16	Jack Russell	Small	NF	Unhealthy	80	16	16
15	Crossbred	Medium	M	Unhealthy	40	32	16
14.5	Crossbred	Small	F	Healthy	0	8	4

## Data Availability

The authors confirm that the datasets analyzed during the study are available from the first author or the corresponding author upon reasonable request.

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
