# Peer review of "Effect of Aging on the Immune Response to Core Vaccines in Senior and Geriatric Dogs"

_vetsci, 2023, doi:10.3390/vetsci10070412_

Round 1

Reviewer 1 Report

The authors present data on the antibody levels of senior and geriatric dogs vaccinated against three core pathogens – CPV-2, CDV and CadV-1.  After comparing age, size, sex, reproductive status, health status, and time between booster shots, the authors recommend that veterinarians consider boosting these animals every 1-2 years, as opposed to the standard practice of every three years.

Recommending that veterinarians consider reducing the interval between vaccinations has its merit.  However, this reviewer has several comments about the paper in its current form.

1.    The paragraph on “human” years (paragraph 2 of the introduction) seems to be unnecessary to this reviewer.  It is an interesting calculation for owners but seems to be a peripheral discussion for a science paper (Supplemental Table 1 also seems not be necessary).

2.     The authors should designate clearly which population is older – geriatric or senior.  This author realizes the size of the dog makes a difference in the aging process, but it would be helpful to know up front that geriatric dogs are considered older than senior dogs.

3.    The authors did a good job of mentioning their study limitations.  This review wonders for some of the limitations, if a review of the dog’s medical records would have provided some of the missing information (vaccination dates, health status, nutritional status, etc). 

There are some minor spelling and grammatical errors that need to be fixed.

Author Response

REVIEWER N. 1

The authors present data on the antibody levels of senior and geriatric dogs vaccinated against three core pathogens – CPV-2, CDV and CadV-1.  After comparing age, size, sex, reproductive status, health status, and time between booster shots, the authors recommend that veterinarians consider boosting these animals every 1-2 years, as opposed to the standard practice of every three years.

Recommending that veterinarians consider reducing the interval between vaccinations has its merit.  However, this reviewer has several comments about the paper in its current form.

R1-C1    The paragraph on “human” years (paragraph 2 of the introduction) seems to be unnecessary to this reviewer.  It is an interesting calculation for owners but seems to be a peripheral discussion for a science paper (Supplemental Table 1 also seems not be necessary).

R1-C2   The authors should designate clearly which population is older – geriatric or senior.  This author realizes the size of the dog makes a difference in the aging process, but it would be helpful to know up front that geriatric dogs are considered older than senior dogs.

R1-A1 and A2. Thank you for your comment. The part related to the comparison with human age was removed from the introduction, while we decided to keep the supplemental Table 1 transforming it in Table 1 (moved it in the main text) and modifying it in order to better show the two groups of interest of our study (senior and geriatric dogs) (lines 62-64)

R1-C3 The authors did a good job of mentioning their study limitations. This review wonders for some of the limitations, if a review of the dog’s medical records would have provided some of the missing information (vaccination dates, health status, nutritional status, etc).

R1-A3. For all dogs we have useful information regarding vaccinations and health status, very useful for interpretation and discussion of the results. We decided to maintain also unhealthy dogs since this condition is really common in the elderly; however, none of those was under treatment with drugs that could interfere with the immune system: for examples, none of the oncological patients was under chemotherapy treatment. This specification was added in the M&M section (line 140-144)

Reviewer 2 Report

Review: [Veterinary Sciences] Manuscript ID: vetsci-2426217

Effect of aging on the immune response to core vaccines in senior and geriatric dogs

The authors describe the effect of core vaccines on the level of antibody response in 2 groups of of older dogs (senior and geriatric) making use of a commercially available dot-ELISA test kit.

The work is useful and of interest to the small animal practitioner and the data presented is significant in my view.

A minor review would however be required before acceptance for publication.

Specific comments:

The English language require editorial attention.

The phrase ‘actual protection’ in line 15 (summary) is disingenuous as actual protection can only be assessed by infectious challenge. Nowhere else in the manuscript is this phrase used and it should be removed here.

The introduction is much too long and should be significantly shortened.

Data that provides the age at which the last vaccine was administ3red would be helpful.

How well does the data that the commercially available kit that was used to assess antibody titers correlate with a gold standard means of determination?

The inclusion and exclusion criteria are much too vague.

A clear definition of age and size groups should be provided.

All the tables and figures are useful except Fig 4 which I think could easily be removed.

Older dogs are more likely to have co-morbid immune suppressing disease and be on immune suppressing drugs. This point needs greater clarification. Potent immune suppressing drugs are commonly used in dogs with neoplasia and the effect of these drugs on protection should be clarified and if possible be reflected in the data presented.

It woud be helpful to have data on the actual incidence of the 3 diseases discussed in the older population of dogs studied. How often is CDV, CAV and CPV actually diagnosed in the population studied. In other words, although and effect of age on the antibody titer ahs been demonstrated, does this effect translate to a real increase in the rate of these infectious diseases in the aged pet population?

There are too many references. Typically around 50 would be regarded as usual in my view.  

The English is understandable and significant edit of language is not required but there are some clear issues with language use that could easily be corrected by a first-language speaker with a veterinary background. 

Author Response

REVIEWER N. 2

The authors describe the effect of core vaccines on the level of antibody response in 2 groups of of older dogs (senior and geriatric) making use of a commercially available dot-ELISA test kit.

The work is useful and of interest to the small animal practitioner and the data presented is significant in my view.

A minor review would however be required before acceptance for publication.

Specific comments:

R2-C1. The English language require editorial attention.

R2-A1. Thank you for your comment, we have paid more attention to English, and we wait for the Editor decision in this matter.

R2-C2. The phrase ‘actual protection’ in line 15 (summary) is disingenuous as actual protection can only be assessed by infectious challenge. Nowhere else in the manuscript is this phrase used and it should be removed here.

R2-A2. The sentence “actual protection” was removed and replaced by “specific serum antibody titers” (line 15).

R2-C3. The introduction is much too long and should be significantly shortened.

R2-A3. Introduction was shortened as requested.

R2-C4. Data that provides the age at which the last vaccine was administered would be helpful.

R2-A4. This information is available for all dogs, and it is different from a dog to another. Most of the dogs were vaccinated annually, many others every 2 or 3 years, and a few only when they were younger. No one was vaccinated for the first time only when old. For our purposes we considered the time since the last vaccination for all dogs, and this information is already present in the manuscript (M&M [lines 145-147] and Results [lines 195-198] sections)

R2-C5. How well does the data that the commercially available kit that was used to assess antibody titers correlate with a gold standard means of determination?

R2-A5. In 2015 Prof. Schultz performed a validation study of VacciCheck (“Schultz, R. A Field and Experimental Trial to Assess the Performance of the ImmunoComb Canine VacciCheck Antibody Test Kit; Biogal Galed Labs: Kibbutz Galed, Israel, 2015”, webpage: https://vaccicheck.com/wp-content/uploads/2014/02/VacciCheck-Performance-Wisconsin.pdf) demonstrating the correlation between this in-clinics test with the gold standard HI and VN. At the Material & Methods’ section we declared that “This test has good specificity and sensitivity for each virus and can be very useful in daily veterinary practice to evaluate the real protection of dogs (and cats) against diseases preventable by core vaccines, as suggested by the international vaccination guidelines and by many other studies [1,40,52–56].”Among those articles, you can find some examples of comparison between this rapid test (VacciCheck) and gold standards, and one of these is our recent study (2022)“Meazzi S. et al. Agreement between In-Clinics and Virus Neutralization Tests in Detecting Antibodies against Canine Distemper Virus (CDV)” (reference 50). Apart these examples, there are many other papers demonstrating the good correlation between VacciCheck and gold standard test, not reported here because out of the topic.

R2-C6. The inclusion and exclusion criteria are much too vague.

R2-A6. All dogs considered senior or geriatric (see new Table 1) were included in this study. As previously explained, we decided to maintain also unhealthy dogs since this condition is really common in the elderly, but none of those was under treatment with drugs that could interfere with the immune system (immunosuppressive or immunostimulant ones): for examples, none of the oncological patients was under chemotherapy treatment. This specification was added in the M&M section (lines 140-145)

R2-C7. A clear definition of age and size groups should be provided.

R2-A8. Thank you for this useful comment. In order to better pinpoint the two groups of interest (senior and geriatric dogs), we decided to modify the supplemental Table S1 and to transform it in Table 1 (moved in the Introduction) (lines 62-64)

R2-C8. All the tables and figures are useful except Fig 4 which I think could easily be removed.

R2-A8. Fig. 4 was removed as suggested by the reviewer

R2-C9. Older dogs are more likely to have co-morbid immune suppressing disease and be on immune suppressing drugs. This point needs greater clarification. Potent immune suppressing drugs are commonly used in dogs with neoplasia and the effect of these drugs on protection should be clarified and if possible be reflected in the data presented.

R2-A9. This is a correct and very important consideration, but the answer is yet present above (R2-A6)

R2-C10. It would be helpful to have data on the actual incidence of the 3 diseases discussed in the older population of dogs studied. How often is CDV, CAV and CPV actually diagnosed in the population studied. In other words, although an effect of age on the antibody titer has been demonstrated, does this effect translate to a real increase in the rate of these infectious diseases in the aged pet population?

R2-A10. To our knowledge there are no specific studies related to epidemiology of these diseases in old canine population neither in Italy nor abroad. However, there are some papers, yet reported in our study, that describe these three infectious diseases in older dogs (references 40-42). Moreover, new variants of CPV-2 (2a, 2b and 2c) are known for being more aggressive and able to infect also adult and old vaccinated dogs (reference 71, 78).

R2-C11. There are too many references. Typically around 50 would be regarded as usual in my view. 

R2-A11. Since the Introduction was shortened, some references have been eliminated. In any case, in our opinion all the references have been used in a correct way, very useful for the reader as well, so we would like to keep them all. In any case, if also the Editor decides that we should reduce them, we can try to better accomplish this request.

Round 2

Reviewer 1 Report

The authors addressed the comments of this reviewer to their satisfaction.

English language is fine.